# Evaluation of Athlete Monitoring Tools across 10 Weeks of Elite Youth Basketball Training: An Explorative Study

**DOI:** 10.3390/sports11020026

**Published:** 2023-01-25

**Authors:** Branson L. Palmer, Grant E. van der Ploeg, Pitre C. Bourdon, Scott R. Butler, Robert G. Crowther

**Affiliations:** 1UniSA: Allied Health & Human Performance, University of South Australia, Adelaide, SA 5000, Australia; 2Alliance for Research in Exercise, Nutrition and Activity (ARENA), University of South Australia, Adelaide, SA 5000, Australia; 3Basketball South Australia, Richmond, SA 5033, Australia

**Keywords:** workload, junior, questionnaire, external, internal

## Abstract

The growth of sport science technology is enabling more sporting teams to implement athlete monitoring practices related to performance testing and load monitoring. Despite the increased emphasis on youth athlete development, the lack of longitudinal athlete monitoring literature in youth athletes is concerning, especially for indoor sports such as basketball. The aim of this study was to evaluate the effectiveness of six different athlete monitoring methods over 10 weeks of youth basketball training. Fourteen state-level youth basketball players (5 males and 9 females; 15.1 ± 1.0 years) completed this study during their pre-competition phase prior to their national basketball tournament. Daily wellness and activity surveys were completed using the *OwnUrGoal* mobile application, along with heart rate (HR) and inertial measurement unit (IMU) recordings at each state training session, and weekly performance testing (3x countermovement jumps [CMJs], and 3x isometric mid-thigh pulls [IMTPs]). All of the athlete monitoring methods demonstrated the coaching staff’s training intent to maintain performance and avoid spikes in workload. Monitoring IMU data combined with PlayerLoad™ data analysis demonstrated more effectiveness for monitoring accumulated load (AL) compared to HR analysis. All six methods of athlete monitoring detected similar trends for all sessions despite small-trivial correlations between each method (Pearson’s correlation: −0.24 < r < 0.28). The use of subjective monitoring questionnaire applications, such as *OwnUrGoal*, is recommended for youth sporting clubs, given its practicability and low-cost. Regular athlete education from coaches and support staff regarding the use of these questionnaires is required to gain the best data.

## 1. Introduction

In reference to youth athletes, the primary aim of elite sporting organisations is to accelerate the growth of their technical and tactical skills [1,2]. While youth sporting clubs and organisations offer this development through training and games at various levels of competition (i.e., recreational and regional), high-performance programs provide a unique opportunity for elite young athletes who aspire to compete at the highest level of competition. However, athletes may fall short of their goals due to their and/or the sporting clubs’ inability to manage the stressors associated with training young athletes [3]. Common issues related to poor management include overuse injuries, repetitive illness, and burnout [4]. Despite the research associated with the negative effects of these issues in youth sports, there is a lack of literature to support the implementation of the best load-monitoring practices to mitigate the risks associated with training this group [5,6]. Hence, research comparing methods of athlete management in youth sports could be extremely beneficial to fostering the development of promising young athletes within high-performance programs.

Athlete monitoring systems assist in describing, planning, and evaluating team training and competitions [7]. Common areas for applying these systems (i.e., monitoring workload, performance, and physical maturity) can involve a vast number of devices, tools, or approaches. Rather than solely monitoring one aspect of youth athlete development, factoring in systems from all areas of athlete monitoring is highly recommended to better understand each athlete’s performance, workload, development, and health [8,9].

Training load monitoring involves assessing the athlete’s short and long-term responses to exercise, typically by analysing their external training load (ETL) and internal training load (ITL) [10]. Solutions for monitoring ETL (e.g., inertial measurement units [IMUs]) are valuable for characterising the work performed during training or competitions; however, it often requires teams to purchase expensive equipment and employ expertise to capture and interpret the data. Alternatively, ITL solutions (e.g., wellness questionnaires and session rating of perceived exertion [sRPE]) are beneficial for evaluating the athletes’ mental and physical responses to training or competition. These internal load measurements are more cost-friendly methods of analysis, but they rely on the athlete’s ability to self-assess their perception of effort and load consistently and honestly [8,11].

Performance monitoring has various applications, including the return to play, developing physical qualities (e.g., speed, muscular power), and understanding an athlete’s movement strategies [12,13]. Although measuring physical qualities, including agility and endurance are important during youth development [14], these performance tests (e.g., the *T*-test and shuttle runs) are practically difficult to implement weekly for athlete monitoring purposes (e.g., longer test durations, conducting tests in a non-fatigue state without compromising skills training, increased equipment and staffing requirements). Fortunately, the countermovement jump (CMJ) and isometric mid-thigh pull (IMTP) are two reliable tests for assessing power characteristics in youth athletes in a time-efficient manner [15,16]. To the authors’ knowledge, exploring the use of the IMTP as a load-monitoring tool has yet to be undertaken in this group. Since sports programs traditionally progress athletes based on their chronological age, non-invasive methods of assessing physical maturity (e.g., age at peak height velocity [PHV]) have been recommended to assist in providing context relating to potential disparities in skeletal growth and chronological age [14]. While there are many logistical limitations to using PHV for grouping athletes based on physical growth (i.e., bio-banding) [17], this method is also cost-friendly to implement. Having data on physical maturity alongside performance testing and training load variables could be extremely helpful for coaches to understand their young athletes’ long-term development.

Research focusing on athlete monitoring has primarily been conducted at the professional and collegiate levels of competition [18,19]. Due to the disparity in team funding and support staff, most of the methods are impractical to use in youth sports. Furthermore, the small amount of athlete monitoring currently implemented in elite youth sporting environments means there is also a lack of literature regarding longitudinal studies comparing the use of multiple athlete monitoring methods. Considering most youth sports teams have limited methods of athlete monitoring available, understanding the utility of each method could provide greater clarity regarding using more cost-effective tools (i.e., questionnaires), as opposed to reliance on high-tech equipment (i.e., IMUs and performance testing).

There are unique environmental factors that impact the monitoring of young basketball players. The smaller competition surface area prevents performance variables such as an athlete’s maximum sprinting velocity from being assessed, compared to field sports that commonly use these metrics [20]. Instead, greater emphasis is placed on performance tests, such as the CMJ and IMTP, to evaluate an athlete’s ability to produce and absorb vertical ground reaction forces. As basketball is played predominantly indoors, global positioning units (GPS) are unable to be implemented. An alternative is using technology such as IMUs to monitor the constant changes in movements during basketball, such as jumping and landing, changes in direction, and rapid accelerations/decelerations. Similar to other sports, wellness and activity questionnaires are commonly implemented in basketball monitoring [5,21]. Despite high-performance youth programs utilising mobile applications, such as *OwnUrGoal*, there is limited research regarding their usefulness compared to other methods of athlete monitoring.

Therefore, the aim of this study was to implement and evaluate state-level youth athletes on the use of athlete monitoring tools (CMJ, IMTP, HR, IMU, activity and wellness questionnaires) across a ten-week pre-competition training period. It was hypothesised that there would be (a) differences in performance testing variables across the training period, (b) a large (−0.5 < r < −0.7) negative correlation between the performance measures and athlete monitoring data, and (c) a moderate correlation (0.3 < r < 0.5) between the internal and external methods of athlete monitoring across the testing period.

## 2. Materials and Methods

### 2.1. Study Design

This longitudinal study consisted of using six methods of athlete monitoring over a 10-week training period. Wellness and activity surveys were completed daily using the *OwnUrGoal* application (Version 1.50.49, OwnUrGoal Pty Ltd., Chandler, Queensland, Australia). All CMJ and IMTP testing was conducted prior to participants commencing their weekly Basketball South Australia (BSA) training sessions at the University of South Australia Magill Sports Centre. The games were located at various indoor high school basketball courts within South Australia. During the BSA training sessions, ETL and HR data were recorded using the Visuallex (VX) Sport IMU (VX Sport Log 340b, Lower Hutt, New Zealand) and VX Sport HR monitors, respectively. The participants wore their regular BSA training and game uniform to all sessions. All of the testing was performed at the same time on each day to attenuate circadian variation. All of the participants and their parent/guardian read and signed an informed consent sheet prior to their study involvement. The study was conducted in accordance with the Declaration of Helsinki and approved by the Human Research Ethics Committee of the University of South Australia (ID: 203498).

### 2.2. Participants

Fourteen state-level youth basketballers (9 females, 5 males; 15.1 ± 1.0 years; 1.80 ± 5.6 m standing height; 0.90 ± 3.4 m sitting height; 70.5 ± 10.8 kg) were recruited for the study based on a sample size power calculation using an effect size of 0.9, alpha error of 0.05, power of 0.9 and 95% coefficient limits (G*Power 3.1.9.4, Dusseldorf, Germany). The inclusion criteria required the participants to be current athletes in a BSA High-Performance program, under 18 years of age, pass stage 1 of the Exercise and Sport Science Australia Adult Exercise Pre-Screening System [22], and free of any musculoskeletal injuries.

### 2.3. Procedures

#### 2.3.1. Demographic Profile and Familiarisation

Each participant received and read an information sheet outlining the purpose of the study and the requirements of their involvement. After informed consent was obtained, the participants then performed a familiarisation session, which included measurement of the participant’s IMTP set-up position (Figure 1), along with performing both the CMJ and IMTP tests. During these performance tests, the participants were fitted with a VX Sports IMU and HR monitor to familiarise themselves with wearing them. All of the participants returned within 1 week to complete their first testing session. The participants’ height (standing and sitting) and mass were measured in weeks 1, 5, and 10 of data collection using a wall-mounted stadiometer (SECA 216, SECA, NY, USA) and Tanita scales (TANITA DR-953 Inner Scan, Tanita, Tokyo, Japan), respectively.

#### 2.3.2. Warm-Up and Rest Intervals

Prior to all of the BSA training sessions, a standardised 10 min warm up that included lunges with rotations, high knees, butt kicks, hip circles, walking scoops, side lunges with squats, and A skips across 15 m was performed. A 3 min rest was provided between each performance test and a 30 s rest was given between the testing efforts [24,25].

#### 2.3.3. Subjective Load Monitoring

The participants were required to complete the daily wellness and activity questionnaires (Figure 2) using the *OwnUrGoal* mobile application. The wellness or activity questionnaires were completed within 30 min of waking up or finishing the activity, respectively [26,27].

The wellness questionnaire (Figure 2a) required participants to rate their levels of muscle soreness, fatigue, sleep quality, mental stress, and motivation on a 1–7 scale (where 1 indicates an excellent score for the respective category, 7 indicates a poor score). Additionally, the participants recorded areas of muscle soreness and hours slept. The questions on *OwnUrGoal* have been modified from Hooper & Mackinnon’s wellness questionnaire [28] by excluding ratings for nutrition and hydration. The participants that reported ‘sick’ were defined as a circumstance in which the participant or their parental guardian felt the athlete was limited or unable to train (flu, virus, etc.) [29].

The activity questionnaire (Figure 2b) required the participant to report details regarding the exercise sessions performed, along with their sRPE, using the modified Borg scale on a 0–10 continuum [30]. The participants were defined as ‘injured’ if they experienced or suffered from pain that restricted them from participating in any basketball training. Injured participants remained a part of *OwnUrGoal* testing, and it was noted that injury was the primary reason for their reduced activity load.

#### 2.3.4. Objective Load Monitoring

Each participant was assigned the same VX Sport IMU (sampling rate 100 Hz) and VX HR monitor (sampling rate 2 Hz) for all testing sessions during the BSA sessions to limit the effect of extraneous variables. The VX Sport IMU was worn on the back, located on top of the 2nd thoracic vertebrae (T2), using the manufacturer’s vest. The VX HR monitor was positioned around their chest and self-adjusted by the participants for firm contact. Maximal HR was assessed for each participant by performing the Yo-Yo intermittent recovery test level 1 (YYIR1) [31] prior to one of the BSA training sessions within 10 weeks. Although the YYIR1 is typically conducted prior to or in the first weeks of monitoring [32], difficulties with commencing the YYIR1 test in a non-fatigued state prevented earlier data collection from occurring. All of the trials were closely monitored by an experienced tester to ensure consistent data collection during testing.

#### 2.3.5. Performance Testing

Prior to each BSA training session, all of the participants performed three maximal effort CMJs and IMTPs on the Advanced Medical Technology (AMTI) portable force platform (1016 × 762 mm) (Advanced Medical Technology Inc., ACP, MA, USA). AccuPower (AccuPower Solutions 2.0.3 Dickinson, ND, USA) and Vicon Nexus (Vicon 2.10.3, Oxford, UK) software was used to capture ground reaction forces, each sampling at 1000 Hz. Prior to all attempts, the force platform was zeroed, and the participants were required to stand still for a minimum of 3 s until they were instructed to commence either the CMJ or the IMTP [33,34].

The CMJ was initiated from a standing position, and CMJ depth was self-selected. The participants held a 1.2 m plastic dowel behind their head, and across their shoulders between the seventh cervical vertebrae and the third thoracic vertebrae during each CMJ (Figure 3) [35,36]. If the participant removed either hand from the dowel or did not land on the force platform, the jump was ruled invalid and repeated. Standardised encouragement of ‘jump as high as possible’ was provided to each participant to assist the participant in maximal performance in each trial. No feedback was provided on the CMJ metrics during the sessions [37], and the techniques for all trials were closely monitored by an experienced operator to ensure safety during maximal testing.

For IMTP testing, a specialised portable rig was constructed and used based on previous studies (Figure 1a) [38,39]. Prior to all attempts, the force platform was zeroed, and the participant’s static weight was determined. After the command “ready”, the participants positioned their feet directly under the bar with their hips and knees angled at exactly 140° (absolute), along with their shoulders directly aligned above the bar (Figure 1b). Hip and knee angles were measured using a hand-held goniometer (Prestige Medical, CA, USA). Previous IMTP protocols have used wrist straps to eliminate grip strength from the test [23,40]. However, due to the practicality of conducting IMTP testing on multiple athletes in a team sport training environment, wrist straps were not used. The participants were instructed to lock the bar to remove slack while also avoiding the application of tension. They were also instructed to ‘pull as hard as possible’ and ‘push the ground away’ to ensure maximal force would be achieved [39,41]. The execution of the IMTP occurred after the command ‘go’ for a 5 s duration. Previous studies have used a countdown method of ‘3, 2, 1, pull’ [23,42]. However, this was not undertaken as the authors believed this could cause youth participants to be impatient and not have a period of static weight. Verbal encouragement was given on all trials, and the technique was closely monitored by an experienced tester to ensure the safety of maximal testing.

#### 2.3.6. Cool Down

At the completion of each session, the participants were given a 10 min standardised cool down, which included walking scoops, high knee pulls and hip circles across 15 m, supine cross-body stretch, and the calf stretch against a wall. A foam rolling was included as an optional component for all participants.

### 2.4. Data Analysis

#### 2.4.1. Demographic Profile

The maturity offset was calculated in Microsoft Excel (Microsoft Corporation, Redmond, WA, USA) using Mirwald et al.’s formula [43]. From this, the predicted age at peak height velocity (PHV) was also calculated by subtracting maturity offset from the participant’s chronological age.

#### 2.4.2. Subjective Load Monitoring

The *OwnUrGoal* data were checked before every BSA training session to ensure compliance. The participants were sent reminders if they had not completed the wellness questionnaire by 9 am and the activity questionnaire within 30 min of BSA training concluding. A wellness score was calculated in *OwnUrGoal* using the total number of points from muscle soreness, sleep quality, stress, motivation, and fatigue. sRPE was also calculated in *OwnUrGoal* using the participant’s session duration multiplied by their rating of perceived exertion. Seven-day load and wellness data were exported weekly as a CSV file.

#### 2.4.3. Objective Load Monitoring

All of the raw accelerometer data (Fx, Fy and Fz) and HR data were exported from VX Sport as separate CSV files. The PlayerLoad™ formula (Equation 1) was used to calculate accumulated load (AL) by combining the changes in movement across all three planes of motion. PlayerLoad™ per minute (PL/min) was calculated by dividing the AL by the session duration (mins) [44].
(1)$$\mathrm{PL}/\min=\frac{\sqrt{[{\left(x_{n}-x_{n-1}\right)}^{2}+{\left(y_{n}-y_{n-1}\right)}^{2}+\left(z_{n}-z_{n-1}{)}^{2}\right]}}{\mathrm{Training}\;\mathrm{duration}\;\left(\mathrm{mins}\right)}$$

Peak HR and mean HR were recorded during each BSA training session as beats per minute (bpm). Training impulse modified (TRIMP mod.) was determined using the following bandwidths: Zone 1 (65–<72% max HR); Zone 2 (72–<79% max HR); Zone 3 (79–<86% max HR); Zone 4 (86 < 93% max HR); and Zone 5 (>93% max HR) [45].

#### 2.4.4. Performance Testing

All of the raw force platform data (Fx, Fy, and Fz) were passed through a fourth-order low-pass Butterworth filter at a cut-off frequency of 20 Hz in the AccuPower Software. The best IMTP (determined by peak Fz ground reaction force [GRF] [N]) and CMJ (determined by jump height [cm] from take-off velocity) from each week were exported from AccuPower as an ACP file. All of the exported data were then transferred to a custom-built Microsoft Excel (Microsoft Corporation, Redmond, WA, USA) document for further analysis [46].

The trapezoidal method (i.e., using take-off velocity) was used to calculate CMJ jump height, which was determined from the Fz GRF [46]. Contact time was calculated using the change in body weight (>5 standard deviations), and flight time was calculated as the time between leaving the ground and landing again using the Fz GRF. The CMJ force–time curve was separated into the following phases: the unweighted phase; the stretching phase; the net impulse phase; the acceleration–propulsion phase; the leaving phase; the propulsion–deceleration phase; the flight phase; and the landing phase (Figure 4) [47]. For each phase, duration (phase length in milliseconds [ms]) and impulse (as newton seconds [N.s]) were calculated. The characteristics of the net impulse phase and factors that directly influence this phase will be examined further, as previous research has suggested net impulse is a significant predictor of CMJ performance [48]. Phase magnitude and impulse were scaled to the participant’s mass and expressed as newtons per kg (N∙kg^−1^) and newton seconds per kg (Ns∙kg^−1^), respectively [25,46].

For the IMTP, absolute peak force (N), relative peak force (N.kg^−1^), peak force duration (ms), rate of force development (RFD, N.s^−1^), and impulse (N.s^−1^) were used as the kinetic variables. The rate of force development was calculated using seven-time zone bands (0–30, 0–50, 0–90, 0–100, 0–150, 0–200, and 0–250 ms) along with peak force RFD [49]. The impulse at peak force 100 and 250 ms was recorded as outcome measures for the IMTP (Figure 5). Furthermore, 100 ms is similar to the ground contact time during sprinting, whilst 250 ms is a ground contact time indicative of jumping activities performed by basketball players [39].

### 2.5. Statistical Analysis

Statistical analyses were conducted using Statistical Product and Service Solutions (SPSS) software (v25, IBM Corp., Armonk, NY, USA). For all of the athlete monitoring variables, weekly means and standard deviations (SD) for each session were calculated, and box plots were used to test for outliers. A Shapiro–Wilk goodness of fit test was used to assess if the data were normally distributed. Linear mixed models were used to assess differences in athlete monitoring variables across each week (week 1 vs. 2 vs. 3 vs. 4 vs. 5 vs. 6 vs. 7 vs. 8 vs. 9 vs. 10). All of the performance testing and VX Sport analyses were performed with *OwnUrGoal* variables as a covariate. Significance level for all *p*-value hypothesis testing was set at *p* < 0.05. Pearson’s bivariate correlation was calculated to assess the relationship between the athlete monitoring methods. The magnitude of the correlation coefficients was evaluated using the following bandwidths: trivial (r^2^ < 0.1); small (0.1 < r^2^ < 0.3); moderate (0.3 < r^2^ < 0.5); large (0.5 < r^2^ < 0.7); very large (0.7 < r^2^ < 0.9), nearly perfect (r^2^ > 0.9), and perfect (r^2^ = 1) [50]. Two-tailed significance level was set as *p* < 0.05.

## 3. Results

### 3.1. Participant Demographics

No differences were observed between weeks 1, 5, and 10 in standing height (week 1 = 1.80 ± 0.05 m; week 5 = 1.80 ± 0.06 m; week 10 = 1.80 ± 0.06 m), sitting height (week 1 = 0.91 ± 0.03 m; week 5 = 0.90 ± 0.03 m; week 10 = 0.90 ± 0.03 m), and body mass (week 1 = 70.5 ± 11.0 kg; week 5 = 71.2 ± 10.8 kg; week 10 = 72.9 ± 11.7 kg). The predicted age of PHV for participants was younger (13.6 ± 0.6 years) compared to their chronological age (15.1 ± 1.0 years).

### 3.2. Pre-Competition Sessions

A detailed outline of all BSA sessions can be found in Table 1. The State Performance Program (SPP) training typically occurred twice per week (one weeknight session and one Saturday session) in preparation for the Australian Junior Classics 2021. These sessions emphasised developing basketball intelligence quotient (IQ) and teamwork through a combination of small-sided games and team scrimmages. Additionally, twelve of the participants were invited to weekly National Performance Program (NPP) training sessions. These weeknight sessions were focused on improving technical skills (e.g., perimeter shooting, ball handling) primarily through small-sided games. All NPP training was conducted across the first nine weeks of the study period. The focus during all of the strength and conditioning sessions was to maintain performance and minimise injury risk.

### 3.3. Subjective Load Monitoring

All of the subjective variables from *OwnUrGoal* can be found in Appendix A. Differences were noticed in the activity reporting, as weekly training loads (WTLs) were both decreased (*p* < 0.05) in weeks 5 and 10 compared to weeks 2, 3, 4 and 6 (Figure 6).

The wellness score, along with muscle soreness, fatigue, sleep quality, and motivation demonstrated no differences (*p* > 0.05) across the 10 weeks (Figure A1 and Figure A2). Higher (*p* < 0.05) hours slept scores were reported in week 10 compared to weeks 4, 7, and 8 and higher (*p* < 0.05) ratings of stress were found in week 7 compared to weeks 3, 8, and 10 (Figure A1 and Appendix A).

### 3.4. Objective Load Monitoring

All HR and IMU raw data can be found in Appendix A. Max HR was lower in week 6 compared to weeks 3, 5, 7, 8 and 10 (Figure A3A). Average HR and all training impulse modified (TRIMP.mod) calculations in weeks 1, 2, 4, and 6 were lower (*p* < 0.05) compared to weeks 3, 5, 7, 8, 9, and 10 (Figure A3B,C).

The average PL/min across the 10 weeks was 78.3 ± 10.7 arbitrary units (AU). PL/min was higher (*p* < 0.05) in week 9 compared to weeks 1, 2, 4–8, and 10 (Figure A3D). Additionally, weeks 2 and 10 were lower (*p* < 0.05) compared to weeks 1, 3, and 8.

### 3.5. Performance Testing

All of the performance testing data can be found in Appendix A. Absolute peak power (A.PP) was greater (*p* < 0.05) in weeks 3, 4, and 10 compared to weeks 5 and 6 (Appendix A). Concentric absolute power (Con. AP) was also greater (*p* < 0.05) in weeks 1, 3, 4, and 10 compared to week 6 (Appendix A). All other outcome variables were similar across the training period. No differences in the CMJ impulse variables occurred across the 10 weeks (Appendix A).

Testing IMTP performance variables demonstrated no differences except for the rate of force development at 250 milliseconds, which was greater (*p* < 0.05) in weeks 5 and 10 compared to week 3 and greater in weeks 5, 8, 9, and 10 compared to week 4 (Appendix A).

## 4. Discussion

The aim of this study was to assess 10 weeks of elite youth pre-competition basketball training using six methods of athlete monitoring (CMJ, IMTP, IMU, HR, wellness and activity questionnaires). This study found that despite small changes in the activity survey and VX Sport accelerometer data, no impactful changes occurred for all other variables. The correlations were trivial-small between performance testing and load monitoring primary measures along with measures of ITL to PlayerLoad™ (Appendix A). To the authors’ knowledge, this is the first longitudinal study comparing multiple methods of performance testing and load monitoring within the same group of state-level youth basketball athletes.

During the 10 weeks of athlete monitoring, the primary focus of the training was to maintain the athlete’s performance and to have all athletes healthy for their national tournament. This coaching concept has been achieved, as demonstrated by the lack of change in the majority of athlete monitoring variables. Hence, a lack of variability occurred in the prescribed training load during the BSA sessions, along with all results for performance testing.

As a result of BSA’s training methodology and goals, most of the wellness and activity questionnaire variables analysed demonstrated little variability between weeks. Due to the lack of variation in ETL (i.e., PlayerLoad^TM^ and WTL) alongside BSA’s intent to reduce outcomes related to muscle soreness and stress, it is difficult to draw conclusions about each variable from the wellness and activity questionnaires. However, both the HR monitors and IMUs were able to detect changes in BSA’s on-court training focus between weeks 5 and 6. Although HR monitors are valuable for recording outcome variables related to movement intensity [51,52], this study highlights how the intensity of individual components of a training session can easily misinterpret the athlete’s demands for the entire session. For example, a coach could run a line sprinting drill, which increases the session max HR; however, the remainder of the session could be low intensity. As PlayerLoad™ is an accumulation effect, it was able to factor in events, such as spot jump shot drills, which are low intensity yet still high volume. Because of this, PlayerLoad™ may be more effective for longitudinal analysis of the youth athlete’s training demands.

In this study, there was a lack of agreement between the subjective variables (i.e., wellness and activity surveys) and all of the objective measures (CMJ, IMTP, HR, IMU) (Appendix A). However, all load monitoring methods, both subjective and objective, identified similar trends between weeks (i.e., little to no variability). *OwnUrGoal* was effective at identifying unique information compared to CMJ, IMTP, HR, and IMU data whilst also displaying similar trends between weeks. Given that training surveys provide the most pragmatic method of longitudinally assessing training demands [53], the use of *OwnUrGoal* or other subjective load monitoring tools is recommended for youth team sports.

In relation to the performance testing conducted in this study, both the CMJ and IMTP tests demonstrated very little week-to-week variability. In addition, there was only a small-moderate agreement between the CMJ and the IMTP (Appendix A). Previous studies have reported a high agreement between these two tests [38,42,54,55]; there are two primary factors to consider for both the lack of variability and agreement between performance tests in this investigation. Firstly, the participants lacked familiarisation with the IMTP test, which was apparent in the variability of body weight prior to commencing their IMTP. Secondly, these performance tests primarily assess separate athletic qualities (i.e., CMJ: reactive strength; IMTP: maximum strength) [39,42]. Therefore, this study has highlighted that these performance tests may monitor an athlete’s neuromuscular fatigue in two separate ways. Using CMJ outcomes is the preferred approach for performance testing in youth athletes due to the participants’ familiarity with the movement pattern.

A limitation of this study was the small sample size of participants. Given the practical difficulties of data collection with high-performance teams, the sample size was outside of the control of the investigators. Taking this limitation into account, linear mixed modelling was conducted to analyse the data sets appropriately. As a result of this common limitation in elite athlete sport science literature [56,57], further research should focus on approaches to individualise athlete monitoring in comparison to the group means. Additionally, the context of the training program designed by the coaching staff did not contain numerous high- and low-intensity training sessions. By analysing an assortment of training demands, the practicality of individual subjective variables alongside the education of participants for completing subjective questionnaires could be studied further.

## 5. Conclusions

In this explorative study, a maintenance in workload for BSA’s high-performance squads was achieved across the 10 weeks of pre-competition training. *OwnUrGoal* provides sufficient information for assisting youth sport coaches and support staff in planning future training sessions without the expense of conducting CMJ, IMTP, HR, or IMU analysis. Future research should investigate the use of individual load-monitoring analysis based on the group data for more individualised exercise prescription, along with the influence of gender and age on youth athlete monitoring practices.

## Figures and Tables

**Figure 1 sports-11-00026-f001:**
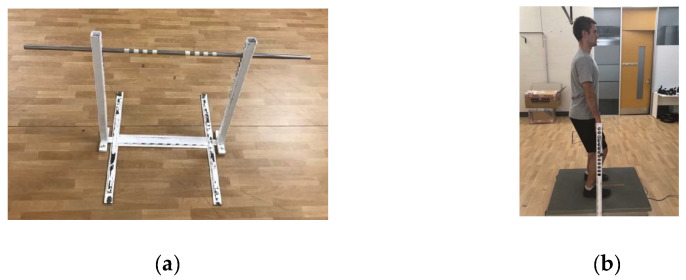
Set-up for Isometric Mid-Thigh Pull testing: (**a**) specialised portable rig; (**b**) participant positioning [23].

**Figure 2 sports-11-00026-f002:**
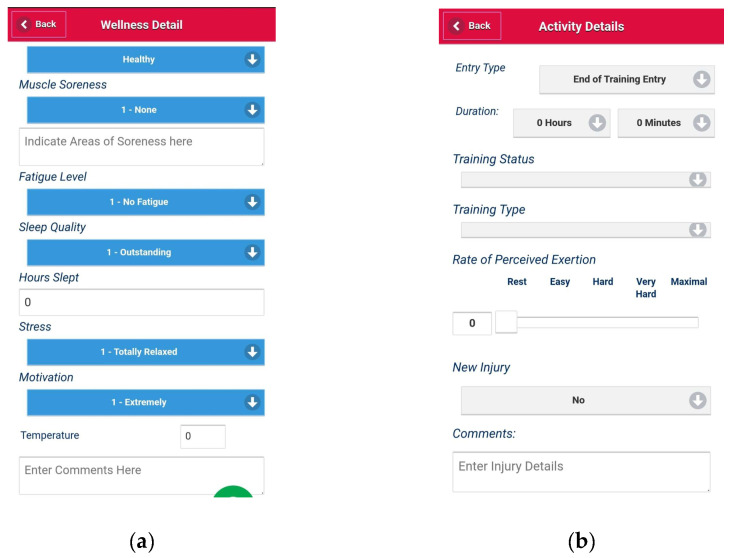
*OwnUrGoal* questionnaires: (**a**) wellness questionnaire; (**b**) activity questionnaire.

**Figure 3 sports-11-00026-f003:**
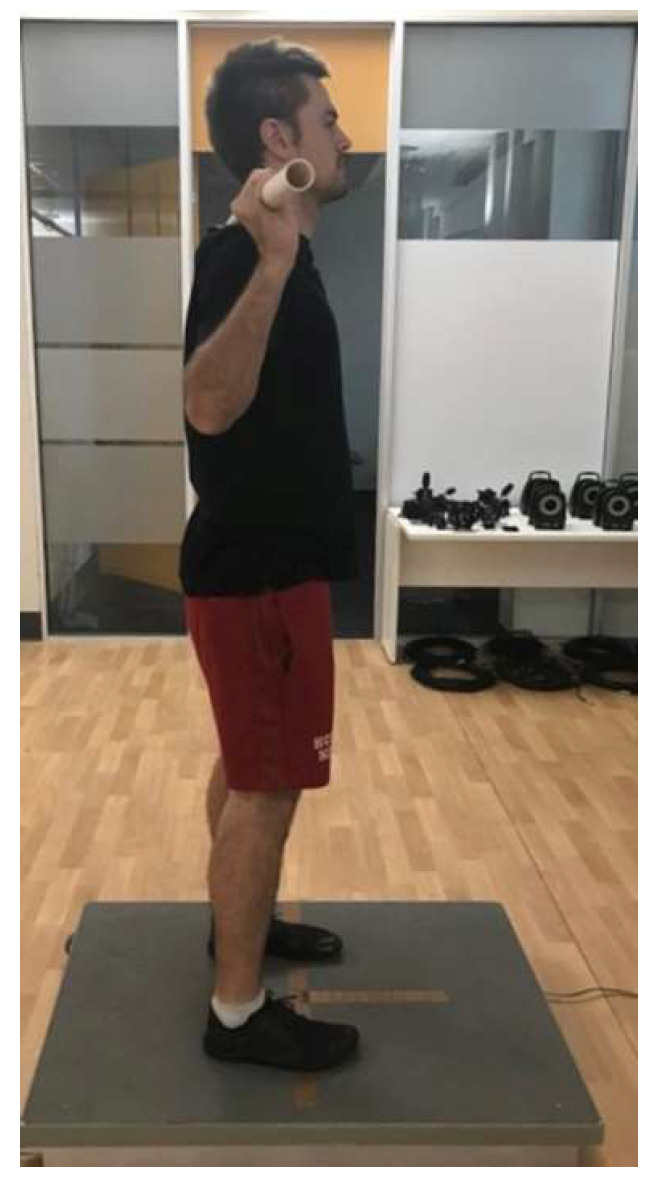
Participant setup on the portable force platform for countermovement jump testing.

**Figure 4 sports-11-00026-f004:**
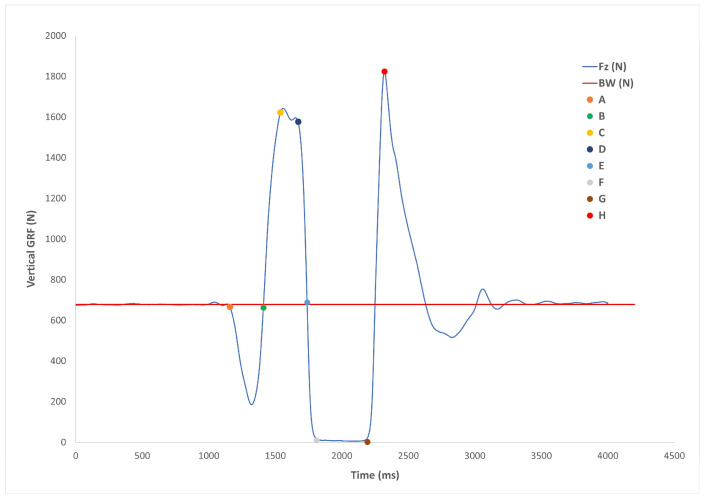
Diagram of the countermovement jump impulse. Point A: Initiation of the jump. Point B: time point where Fz returns to system weight. Point C: Peak negative displacement of the jumper’s centre of mass, as well as the end of the eccentric phase. Point D: The end of net impulse. Point E: Fz falls below system weight and peak velocity of the jumper’s centre of mass. Point F: Take off. Point G: Landing. Points A to B: unweighted phase. Points B to C: stretching phase. Points C to D: net impulse. Points C to E: acceleration-propulsion phase. Points D to E: leaving phase. Points E to F: deceleration-propulsion phase [46].

**Figure 5 sports-11-00026-f005:**
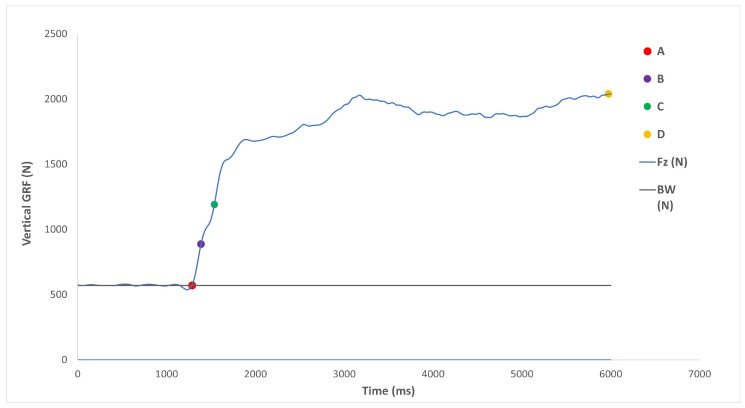
Diagram of the isometric mid-thigh pull impulse. A: onset of IMTP, B: onset to 100 ms, C: onset to 250 ms and D: peak force.

**Figure 6 sports-11-00026-f006:**
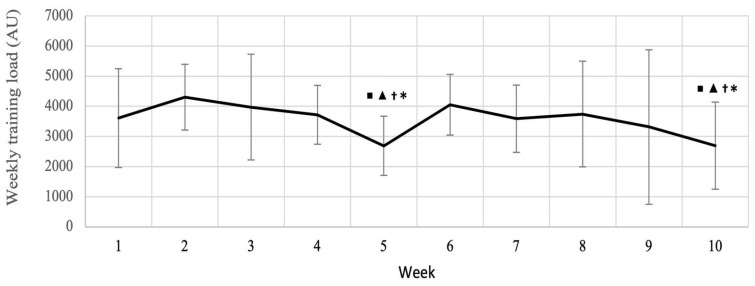
Weekly average loads for all reported trainings and competitions over 10 weeks. ▪ < 0.05 vs. week 2; ▲ < 0.05 vs. week 3; † < 0.05 vs. week 4; * < 0.05 vs. week 6.

**Table 1 sports-11-00026-t001:** Outline of the training timetable during 10-weeks of pre-competition training.

Training Squad	Week 1	Week 2	Week 3	Week 4	Week 5	Week 6	Week 7	Week 8	Week 9	Week 10
Under 16 Girls	2x SPP 1x NPP	2x SPP 1x NPP	2x SPP 1x NPP	2x SPP 1x NPP	1x SPP 1x NPP	1x SPP 1x NPP	2x SPP 1x NPP	2x SPP 1x NPP	1x NPP	1x SPP
Under 16 Boys										
Under 18 Girls					1x G 1x NPP	1x G 1x NPP		1x G 1x NPP	1x G 1x NPP	
Under 18 Boys										
**On-Court** **Training Focus**	Skills development, small-sided games, and scrimmages (full and half court)					Decreased full court running and player contact (i.e., more half court small-sided games and team scrimmages)				

SPP = State Performance Program training; NPP = National Performance Program training; G = game.

## Data Availability

The data presented in this study are available upon reasonable request from the corresponding author.

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
