# Peer review of "Evaluation of Athlete Monitoring Tools across 10 Weeks of Elite Youth Basketball Training: An Explorative Study"

_sports, 2023, doi:10.3390/sports11020026_

Round 1
Reviewer 1 Report
I took great pleasure in reading this study, which investigating the athlete tracking system in this age group. I congratulate the authors for working on this age group, and particularly in an area like basketball where there is little study of talent development. Study is overall good but Just I have a few concern which I write below.
Line 67: what about other performance indicators such as as you mentioned speed, agility or endurance ? why authors choose to mentioned only from power monitoring with aforementioned measurements ?
Line 120: based on which study did you calculate your sample size ?
Line 159: Why auyhors excluded nutrition and hydration parameters that may increase each training session performance which in turn affect avorall training adaptations ? I just ask
Line 182: why authors did not measure other physical performance parameters such as agility which is far most important parameters during these years to differentiate from peers ?
Line 231: please dont forget to finish the “track changes” option before resubmission.
In the results section, i tıhink it can be a good option to present real P values rather than p> and p<…
In the whole disccusion, I just realize that there is poor researches that compared with each other, authors should add more other study’s data to the discussion section.
Reviewer 2 Report
Dear Authors,
I would like to express my gratitude regarding the opportunity to review this manuscript.
The study topic is interesting, at this stage the manuscript requires considerable improvements. Below comments and suggestions with line indication:
2-4 – Please review the upper and lowercase in the title. Please consider journal template and instructions for authors not only in the title, but throughout the manuscript.
31 – Please consider lowercase.
99 – Please include manufacturer country.
109 - Please describe all the procedures in detail. For example, where were the evaluations performed? Conditions (temperature, humidity?) Previous nutrition? Clothes during data collection? Who collected the data, training, and experience? Time of data collection (circadian effect)? All details should be considered and detailed.
109 – Ethical code should be provided (content in lines 431-433), and Helsinki Declaration indicated.
118 – Please indicated subjects characteristics.
124 – Please include exclusion criteria.
153 – Please insert end point.
231 – Please remove word track change.
268 – Please correct “-1” to above the text throughout the manuscript.
269 – Please improve the figure quality.
281 - Please remove word track change.
286 - Please improve the figure quality.
303-304 – Please revise the space considering the journal template.
323 – Please consider journal template and instructions for authors regarding the table format.
329 – Please improve figure quality.
355 – The discussion section requires improvements, is too short. For example, the sample had male and female players, this should be addressed in this section, among other issues to be analyzed with bibliographic references support.
444 – Appendix A and B should be considered in the results section and the figures quality should be improved.
460 – All references format should be corrected considering the journal template and instructions for authors.
Please carefully revise all manuscript and consider English improvement.
Reviewer 3 Report
General comments
This manuscript aims at evaluating the effectiveness of six different athlete monitoring methods over 10 weeks of youth basketball training. In spite of some specific issues detailed below, authors manage to fulfill sufficiently their aim.
Specific comments
How was sample size chosen? I mean starting from which measures coming from some pilot study or literature?
Relevant missing ref:
https://pubmed.ncbi.nlm.nih.gov/32560400/
Minor comments
(line 23 and elsewhere throughout MS, as well) Please, do not start sentences with acronyms;
(l327) no “Appendix G” found (throughout sports-2110227-peer-review-v1.pdf and sports-2110227-supplementary.docx files): sure about naming?
(l344) … 10 (Figure B2). Additionally…
Round 2
Reviewer 2 Report
Dear Authors,
The manuscript improved during the review process, congratulations.
At this stage (v2) the document requires improvements of some details, below with line indication:
5-10 - Please carefully review the journal template and journal instructions for authors and consider these in all manuscript. Please check last published articles in the journal (https://www.mdpi.com/journal/sports). For example, in these particular pages, zip code is missing.
31 – Please remove end point.
127 – Please indicate institutional ethical submission and that participants or guardians provide written consent (placed in 452-453).
241 – “et al.” is missing, please correct.
265 – Please confirm if no more than one space.
343 – “explorative study” – Please consider adding to the title of the manuscript.
493 - All references format should be corrected considering the journal template and instructions for authors. Por example the volume should be in italic (without v) and afterwards number between parentheses. “pp.” should be removed. Again, please review the journal template and journal instructions for authors and consider these in all manuscript. Please check last published articles in the journal (https://www.mdpi.com/journal/sports).
It is very important to perform a careful final reading after considering the suggestions, with special attention to details (document format according to the journal, possible improvement of English, correction of small details).
Author Response
Please see the attachment.
We've applied some updates from additional comments as well, however the next version will include all images that meet the requirements (submitted in a ZIP file), along with any necessary changes that we become aware of in the meantime.
